# Metabolic Footprinting of Microbial Systems Based on Comprehensive In Silico Predictions of MS/MS Relevant Data

**DOI:** 10.3390/metabo12030257

**Published:** 2022-03-17

**Authors:** Alexander Reiter, Jian Asgari, Wolfgang Wiechert, Marco Oldiges

**Affiliations:** 1Institute of Bio- and Geosciences, IBG-1: Biotechnology, Forschungszentrum Jülich GmbH, 52425 Jülich, Germany; a.reiter@fz-juelich.de (A.R.); jian.asgari@rwth-aachen.de (J.A.); w.wiechert@fz-juelich.de (W.W.); 2Institute of Biotechnology, RWTH Aachen University, 52062 Aachen, Germany; 3Computational Systems Biotechnology, RWTH Aachen University, 52062 Aachen, Germany

**Keywords:** metabolomics, dilute and shoot, flow-injection analysis, mass spectrometry, prediction, bioprocess development, method development, automation, digitalisation, yeast extract

## Abstract

Metabolic footprinting represents a holistic approach to gathering large-scale metabolomic information of a given biological system and is, therefore, a driving force for systems biology and bioprocess development. The ongoing development of automated cultivation platforms increases the need for a comprehensive and rapid profiling tool to cope with the cultivation throughput. In this study, we implemented a workflow to provide and select relevant metabolite information from a genome-scale model to automatically build an organism-specific comprehensive metabolome analysis method. Based on in-house literature and predicted metabolite information, the deduced metabolite set was distributed in stackable methods for a chromatography-free dilute and shoot flow-injection analysis multiple-reaction monitoring profiling approach. The workflow was used to create a method specific for *Saccharomyces cerevisiae*, covering 252 metabolites with 7 min/sample. The method was validated with a commercially available yeast metabolome standard, identifying up to 74.2% of the listed metabolites. As a first case study, three commercially available yeast extracts were screened with 118 metabolites passing quality control thresholds for statistical analysis, allowing to identify discriminating metabolites. The presented methodology provides metabolite screening in a time-optimised way by scaling analysis time to metabolite coverage and is open to other microbial systems simply starting from genome-scale model information.

## 1. Introduction

The metabolism of microorganisms is a complex and dynamically regulated system of biochemical reactions. These enzyme-catalysed reactions provide the necessary intermediates for cell growth, energy conversion, product formation, and the possibility for adaptation to environmental changes. The interactions of genome, transcriptome, proteome, and metabolome result in certain metabolic sets and lead to the fluxome as part of this multilayer system [1,2]. As a metabolic image of the cell state, the metabolome allows the investigation of environmental changes in the organism based on a given genetic set [3]. This allowed metabolomics to emerge as a valuable tool in life science applications, such as systems biology [4,5,6], synthetic biology [7,8], and clinical research [9,10,11].

In microbial research, metabolomics can be utilised as a tool for a wide range of questions, e.g., enzyme function analysis or discovery [12], gene function relationship [13,14], pathway identification [15], and metabolic flux analysis [16,17]. The findings of these areas can be transferred to industrial biotechnology [18,19,20], where metabolomics additionally supports, e.g., microbial strain engineering [21,22,23,24], strain characterization [25,26,27], bioprocess monitoring [28,29,30], and optimization [17,31,32].

With the ongoing development of molecular biology and metabolic engineering tools, large microbial producer strain libraries can be generated in a short amount of time [33,34,35]. Subsequently, these libraries are characterised in miniaturised or small-scale cultivation systems [36] with high throughput, which is often supported by laboratory automation technology [37]. While these systems accelerate the cultivation throughput and could allow for time-resolved sampling [38], they usually lack the option for a comprehensive individual phenotypic characterization in terms of the full spectrum of metabolic (by-)products, providing biological insight for strain and bioprocess engineering. Thus, the need for comprehensive high-throughput metabolomics is an obvious consequence to generate hypotheses on molecular mechanisms and metabolic regulation to support further metabolic and bioprocess engineering [39].

Mass spectrometry (MS) is widely established as the detector of choice for metabolomic studies [40,41]. Based on the selected mass analyzer technology and benefits of the analyzing modes, MS allows the identification or quantification of small molecules in untargeted or targeted metabolomic studies [42,43]. A quadrupole-time-of-flight (QqToF) analyzer merges the capabilities of quadrupole precursor selection and fragmentation with ToF high mass resolution and accuracy, making it a valuable tool for metabolite identification and allowing the quantification of metabolites in product ion scan (PI) mode. Still, if operated with the same dwell and cycle time, a triple-quadrupole (QqQ) in multiple-reaction monitoring (MRM) generally shows a higher quantitative precision and sensitivity. In addition, a low mass resolution QqQ is especially suited for high-throughput applications because of its robustness and waiver of in-between batch mass calibration with a calibrant delivery system.

Nevertheless, the bottleneck of a targeted tool such as MRM for large-scale metabolomic studies is the limitation in metabolite coverage. The metabolite-specific information for fragmentation pattern, ionization mode, and other information needs to be known or derived from physical properties to create a targeted MS/MS assay. The gold standard to gather such information is the manual parameter optimization with metabolite standards to acquire not only the MS/MS pattern but also information about ionization mode and optimal mass analyzer settings. However, only a small part of the microbial metabolome is available as reference standards with substantial costs. Literature can also be a fruitful source of MS/MS data information [44,45,46,47], barring some uncertainties with respect to method transfer from different MS devices and vendors.

The application of a QqQ and the utilization of its MRM mode for global scale metabolomic studies was already demonstrated by applying incremental methods to determine a set of optimization parameters of relevant metabolites prior to actual analysis [48,49,50]. While such approaches can cope with the challenge of instrumental parameter selection, they are often limited by the acquirable throughput. Independent of the method used, the metabolite or parameter information is acquired during or in between the measurements, resulting in necessary additional analysis runs. In addition, such incremental methods are tailored for the respective samples measured and need to be repeated for every new sample type.

In silico obtained MS/MS spectra might allow closing the gap between metabolite standards availability and the vast range of naturally occurring metabolites [51,52]. Collision-induced fragmentation modelling allows the prediction of metabolite fragment spectra based on known datasets in a machine learning approach [53,54,55]. While generally used for metabolite identification [56,57], in silico spectra might be promising sources for mass transitions of metabolites in a targeted approach [44].

MS methods are frequently used in conjunction with liquid-chromatography (LC) methods, separating analytes from buffer molecules, salts, and other analytes, thereby minimizing matrix effects and charge competition leading to improved overall signal intensity [58,59]. In a large-scale targeted approach, the limiting cycle time of a quadrupole could be overcome by scheduled MRM modes [60]. However, with increasing sample numbers, the run time for LC becomes a critical step limiting the overall throughput, which represents a clear disadvantage. This holds especially true in a setting with a high number of metabolites and high throughput.

In this study, we present a global LC-free dilute and shoot (DS) flow-injection analysis (FIA) QqQ method for semitargeted metabolomics. Based on a compound-specific set of manually optimised in-house and literature MS data, in silico-predicted product ion spectra and pK_a_ values were validated and subsequently used to assemble metabolite-specific MS/MS mass transitions, collision energies, and the ionization mode.

By applying the information of the KEGG, PubChem, and ChEMBL database, an ion library of 19,948 mass transitions for 5110 metabolites was created. Mass transition selection of individual metabolites was based on selecting the corresponding genome model of the organisms, considering isobaric and amphoteric molecules, as well as evaluating signal intensity. To provide sufficient dwell time and analytical precision, the mass transitions were distributed in packages with up to 40 transitions for positive or negative ionization modes. In the LC-free FIA approach, we reduced the analysis time of a single injection to 1 min/package. To encounter matrix effects and charge competition, we applied high dilution factors in a DS approach and improved ionization by an organic modifier. The method was validated with a commercially available *Pichia pastoris* (PPA) cell extract. Subsequently, it was applied to different types of commercially available yeast extract products (YE) as a model system for a complex microbial sample material consisting of several hundred metabolic compounds. Utilizing a QqQ for detection allowed the precise determination of metabolite peak areas for multivariate discrimination analysis and identification of the most discriminating metabolites in the YE products. The approach is not limited to yeast metabolites and can be transferred to any microbial system simply starting with the corresponding genome-scale model.

## 2. Results

The extension of a targeted QqQ method to a broader scope of metabolites requires detailed metabolite information in the form of molecular weight, fragmentation pattern for given collision energy, and possible ionization states for every compound. Such information was obtained from manually optimised MS parameter studies for 146 metabolites by direct injection (DI) of standard solutions with a QqToF in PI mode. Moreover, detailed MS/MS-specific information in the form of MRM parameters for additional 272 metabolites could be obtained from the literature [44,61,62,63,64,65,66,67,68], expanding the experimental parameter database to 418 metabolites (provided in the repository). Nevertheless, a substantial part of the metabolome was not covered, demanding the application of in silico approaches to close this gap.

### 2.1. Prediction and Validation of pKa Values to Select Ionization Mode

Selecting the appropriate ionization mode for a given metabolite is crucial for soft ionization techniques such as electrospray ionization (ESI). Amine groups may be protonated for ionization in positive mode, while carboxyl groups can be deprotonated to allow for ionization in negative mode. Thus, the pK_a_ value of the entire molecule could be used to select the appropriate ionization mode in the method development. Unfortunately, the availability of experimentally determined pK_a_ values is limited, mainly because of the same reasons as for the MS/MS spectra. The key to success for pK_a_ prediction is the information about pK_a_ values for the acidic and basic groups in the molecule.

The Kyoto Encyclopedia of Genes and Genomes (KEGG) [69,70,71] was used as a source of biological pathway and metabolite information. The unique compound identifier (ID) of the KEGG small molecule category was selected as an unambiguous metabolite compound identifier during database creation and subsequently filtered by additional queries (Figure 1a). The benefit of using KEGG as the starting point for database creation is its versatile connectivity to genome-scale models. However, KEGG does contain only sparse information on the chemical compound structure or relevant parameters for MS analysis, demanding further connection to other databases using the KEGG ID for compound tracing. Therefore, a four-step procedure was conducted to (i) get the structural metabolite or substance information; (ii) get the neutrally charged metabolite or compound identifier (ID); (iii) link the neutrally charged metabolite to a pK_a_ database; (iv) obtain pK_a_ values for the metabolite.

(i) The small molecule class of KEGG provided 18,749 compound identifiers with 18,598 corresponding PubChem substance identifiers (SIDs). (ii) PubChem [72] and the PubChem SIDs were used to acquire the standardised notation of the molecule structures and the linked compound identifier (CID) to avoid charged molecule forms or tautomers. (iii) Of the 16,482 neutrally charged metabolites with a CID, 7619 metabolites were linked with a ChEMBL ID. ChEMBL [73,74] was the key to acquiring predicted pK_a_ values of functional groups for further decision making on ionization mode selection. (iv) Overall, for 5676 metabolites, the predicted pK_a_ values for acidic or basic functional groups could be provided (available in the repository). The ChEMBL classification and distribution of predicted pK_a_ values over the pH range are shown in Figure 1b.

The pK_a_ values of the corresponding category represent the predicted logarithmic acid constant of the acidic or basic groups individually in the case of multiple functional groups in the molecule and not their overall pK_a_ values. Based on ChEMBL, the remaining KEGG molecules are classified in 2004 acids, 1029 bases, 686 zwitterions (referred to as amphoterics), and 3520 neutral molecules. A graphical representation of the ChEMBL classification for acids, bases, amphoteric, and neutral molecules is given in Appendix A. Acids consist of molecules with strong and weak proton donors, e.g., phosphate and carboxylic groups with either acidic pK_a_ values < 6.5 (only acidic side groups, e.g., uridine triphosphate) or acidic pK_a_ values < 6.5 and basic pK_a_ values < 8.5 (acidic and basic side groups with overall strong acidic behaviour, e.g., adenosine triphosphate). Classified bases consist mostly of organic amines with basic pK_a_ values> 8.5 (only basic side groups, e.g., ammonia) or basic pK_a_ values > 8.5 and acidic pK_a_ values > 6.5 (strong basic behaviour, e.g., kanamycin A). Amphoteric molecules contain acidic pK_a_ values < 6.5 and basic pK_a_ values > 8.5 (acidic and basic side groups, e.g., amino acids). Neutrally labelled molecules contain either acidic pK_a_ values > 6.5 (weak acidic behaviour, e.g., tricine), basic pK_a_ values < 8.5 (weak basic behaviour, e.g., proflavine), or acidic pK_a_ values > 6.5 and basic pK_a_ values < 8.5 (molecules not covered by classification so far, e.g., pyridoxine).

The validation of predicted ChEMBL pK_a_ values was conducted with experimental pK_a_ values from the DataWarrior pK_a_ value set [75] in a quantitative structure–activity relationship (QSAR)-ready form or QSAR-SMILES strings provided in [76] for 374 matched metabolites. Based on the predicted pK_a_ values for acidic and basic groups in the molecules, metabolites were categorised in acids (only acidic pK_a_), bases (only basic pK_a_), and amphoteric species (acidic and basic pK_a_). Evaluation of prediction was performed based on linearity by linear regression and absolute deviation for each individual compound (Figure 2a,b).

The correlation analysis (Figure 2a) between 374 predicted and experimental pK_a_ values of metabolic compounds displays an overall linear relationship which is close to the bisecting angle with a coefficient of determination r^2^ = 0.9301. The absolute pK_a_ deviation (Figure 2b) of acids and bases is similar and in between approximately ±3 pK_a_ units, while the amphoteric species show deviations up to ±4 pK_a_ units, which might be sufficient for simple ionization mode selection.

Overall, the predicted pK_a_ values for the functional groups of the metabolite molecules showed sufficient accuracy for simple ionization mode selection and additionally provided chemical ionization permission by providing pK_a_ values of acidic and basic functional groups in the molecule. This allows allocating, for example, molecules with an acidic pK_a_ and without a pK_a_ of basic groups to the negative ionization mode. For molecules classified as amphoteric (pK_a,acidic_ < 6 and pK_a,basic_ > 8.5) or neutrally labeled metabolites with weak acidic and basic properties (pK_a,acidic_ > 6 and pK_a,basic_ < 8.5), ionization mode selection will be combined with mass transition selection to avoid isobaric convolution patterns if possible. Isobaric molecules show identical precursor and product ion m z^−1^.

### 2.2. Prediction and Validation of Metabolite Fragmentation Pattern

For a given metabolite, the selection of the ionization mode is generally followed by the selection of a suitable mass transition to provide high MS sensitivity and selectivity in a targeted and semitargeted approach with a priori information. Since metabolite reference standards are often not available, metabolite fragmentation patterns can be predicted. The PubChem data were not only a valuable source of information for pK_a_ value estimation and selection of ionization mode but also provided the simplified molecular input line entry specification (SMILES) strings for the compounds. SMILES strings were obtained from PubChem data and subsequently used for fragment spectra prediction with the competitive fragmentation modelling (CFM) tool [53,54] for three collision energies (CE) of ±10 V, ±20 V, and ±40 V.

The use of predicted MS/MS spectra might bear a risk due to the modelling error of the CFM-ID fragment prediction model itself, the training set sizes, and the origin used. To minimize misconceptions about the model, the MS/MS prediction was revalidated with experimental in-house spectra of available reference standards. Experimental metabolite fragmentation spectra were acquired by DI of metabolite standard solutions on a QqToF with rolling collision energy ramps in PI mode. Collision energy ramps ranged from -130 to 0 V in negative mode and 0 to 130 V in positive mode. PI spectra at ±10 V, ±20 V, and ±40 V of 61 metabolites were used to evaluate Recall (R), Weighted Recall (WR), Precision (P), Weighted Precision (WP), and Jaccard scores (J), described in [53], with their corresponding prediction (Figure 3b–g). An example is displayed for tyrosine (KEGG ID C00082) in positive ionization mode with +20 V collision energy (Figure 3a). Further examples are provided in Appendix A. The results are averaged over all predictions (Figure 3b), per ionization mode (Figure 3c,d), as well as collision energy (Figure 3e–g).

Weighted as well as unweighted metrics were used for evaluation [53]. R and P as unweighted metrics favour matched high and low-intensity signals equally. This allows to generally evaluate the accuracy of prediction in both directions, including low-intensity signals to avoid overemphasizing single signals. For the example given in Figure 3a, in total, 8 MS/MS fragment signals were measured, while the CFM-ID tool predicted 18 potential fragments. Thus, six out of eight signals in the measured spectrum can be found in the prediction resulting in recall (R) = 75%. Vice versa, 6 out of 18 signals in the prediction can be found in the measured spectrum resulting in precision (P) = 33.3%. Since R and P might be misleading in the case of low-intensity signals such as noise, weighted recall (WR) and weighted precision (WP) were also evaluated. WR and WP represent weighted R and P, which additionally take the signal intensity into account. This favorus high-intensity signals, which might provide a better indication of how much a spectrum has been matched. Therefore, a relatively high-intensity signal in the measurement (indicated in Figure 3a, blue spectrum) with a match in the predicted spectrum (red spectrum) has a high impact on WR. The Jaccard score (J) is an unweighted indicator for overall compliance between measurement and prediction, including low-intensity signals such as noise. The conjunction (six matching signals) divided by the disjunction (20 signals) of measured and predicted signals results in a Jaccard score (J) of 30%.

The percentage of the total peak intensity in the measured spectrum with a matched peak in the predicted spectrum (WR) shows a maximum of 91% for 10 V (Figure 3e) and a minimum of 78% for 40 V (Figure 3g), indicating that the measured fragment response is most probably present in the predicted fragment spectra. The decrease in WR by the increase in collision energy (Figure 3e–g) might be due to subsequent fragmentation, which leads to an increase in fragment variability and a decrease in high response signals.

On the contrary, the percentage of the total peak intensity in the predicted spectrum with a matching peak in the measured spectrum (WP) between 63–66% (Figure 3b–g) is slightly lower than the corresponding WR. This indicates additional relevant fragment responses in the prediction which are not present in the measured fragment spectra. This may lead to a false positive mass transition selection if the predicted response or mass transition might be the most dominant and therefore selected. Subsequently, this would result in false negative metabolite identification.

The percentage of peaks in the measured spectrum that have a matching peak in the predicted spectrum (R) is 80% for 10 V (Figure 3e) and 60% for 40 V (Figure 3g), showing a similar ranking to the corresponding WR. With the inclusion of small responses, the drop from WR to R describes additional predicted fragments with lower intensity. Similar to R, the percentage of peaks in the predicted spectrum that have a matching peak in the measured spectrum (P) 27% for 10 V (Figure 3e) and 38% for 40 V (Figure 3g) decreases accordingly by considering low abundant fragments. The conjunction divided by the disjunction of measured and predicted fragments (J) of 24–29% (Figure 3b–g) stresses the challenge of low abundant fragment prediction even further.

The missing coverage or false prediction of small, abundant fragments by MS/MS prediction displays a systematic limitation of prediction and, in comparison, to manual parameter optimization. While the low abundant predicted fragments might actually occur in the experimental fragmentation process, they could be below detector limit or suppressed by charge competition. On the contrary, the prediction might be inaccurate for the corresponding metabolites. Nevertheless, small, abundant fragments either in predicted or measured MS/MS spectra might be negligible for mass transition selection to avoid low ion intensities. Since the WR and WP range from 65–85% for all predictions (Figure 3b), sufficient prediction capabilities for relatively high-intensity fragments for mass transition selection are clearly demonstrated. R, P, and J were included to provide a comprehensive evaluation of the prediction and to allow a comparison to the original study. Overall, the results confirm the predictive capabilities of the model stated in [53] to a great extent with an independent in-house MS/MS dataset.

### 2.3. Automated MS/MS Method Assembly

The predictions for valid metabolite MS/MS fragment spectra and pK_a_ values allows us to identify suitable mass transitions, collision energies, and ionization modes for every metabolite. Based on a *Saccharomyces cerevisiae* genome model (SCE) of KEGG, metabolites with predicted fragment spectra and pK_a_ values were selected from the database and subsequently used to create an organism-specific MS/MS method for 252 metabolites. Although the KEGG metabolome of SCE lists up to 724 metabolites, only 252 of those metabolites were linked to a pK_a_ value of at least one ionizable group. The difference of 472 metabolites consists of molecules lacking available pK_a_ data either by a missing ionizable group (e.g., carboxyl, phosphate, amine) or a cross-referenced pK_a_ value for the groups in ChEMBL (e.g., urea). While the soft ionization of metabolites without at least one acidic or basic group is not applicable, no allocation to positive or negative ionization mode is performed for such compounds. Thus, the 252 fully described metabolites were selected for ionization mode allocation and mass transition selection.

Further, metabolites were classified based on mass transition origin, selected ionization mode based on pK_a_ values, and isobaric convolution pattern (Figure 4).

The evaluation of mass transition origin shows 81 in-house, 76 literature, and 95 predicted mass transitions (Figure 4a) clearly demonstrates the importance of in silico predictions. While 62% of the mass transition are based on experimentally determined values from either in-house or literature, 38% were predicted data, demonstrating the current lack of detailed metabolite information by experimentation and underlines the demand for alternative information sources such as in silico predictions. When looking at the distribution of in-house, literature, and predicted mass transitions over the m z^−1^ axis, they were nearly evenly distributed over the whole range (Figure 4a).

Concerning the distribution of the metabolites to the ionization modes, 96 metabolites are in positive and 156 metabolites in negative mode (Figure 4b), which could indicate a bias to functional groups such as phosphate of carboxyl groups in the metabolite set. The metabolites for positive and negative ionization are broadly distributed across the m z^−1^ axis, with most metabolites being within approximately 150 and 250 (Figure 4e).

With regard to the uniqueness of mass transitions, 218 metabolites showed this important feature, while 34 signals were convoluted, i.e., nonunique mass transitions (Figure 4c). Unlike unique mass transitions, convoluted mass transitions were especially present in the region of 100–250 m z^−1^ (Figure 4f), which could be a consequence of the overall higher count of metabolites with similar structure and size in this region. Although convoluted mass transitions are labelled, the accuracy of such allocation depends on the predicted part of the mass transitions and the accuracy of the utilised prediction model.

Overall, the distribution of precursor m z^−1^ is skewed, showing more metabolites in the range up to 400 m z^−1^, with approximately 50% of all metabolites being below 200 m z^−1^ representing the dominating set of small molecules in SCE.

### 2.4. Validation

The automatically generated method covering the 252 yeast metabolites was evaluated by analyzing a commercially available *Pichia pastoris* (PPA) yeast extract metabolome standard with 94 routinely identified metabolites listed in the certificate of analysis. The metabolome standard was diluted at 1:10^3^ with 50% MeOH (*v/v*) and analysed with DS-FIA-MS/MS in six technical replicates. The selection of the mobile phase was conducted based on an earlier study, including absolute quantification of amino acids in highly salt-loaded and additionally buffered cultivation media with DS-FIA-MS/MS [77]. Based on this targeted LC-MS/MS method, the mobile phase for positive ionization mode (5% MeOH and 5% acetic acid in H_2_O) was optimised utilizing an organic modifier in the eluent. The mobile phase selection for negative ionization mode was based on initial experiments (data not shown). Different modifiers encountered in LC-MS/MS, such as NH_3_OH and MeOH, were used as mobile phase additives in analysing a yeast metabolome standard. A fully organic MeOH phase showed the broadest metabolite coverage and lowest analytical error in the DS-FIA-MS/MS approach. The 252 mass transitions were distributed in three methods with positive ionization mode and four methods with negative ionisation mode, resulting in a total analysis time of 7 min/sample with an average of 36 MS/MS transitions per package. The method evaluation by metabolite verification is displayed in Figure 5.

For consistent nomenclature of metabolic compounds, all metabolite names listed in the certificate of analysis were translated to KEGG annotation (see Sheet 1 in ESM2). Surprisingly, only 89 of 94 metabolites are annotated in the PPA extract, based on the KEGG metabolome as a result of the entire list of genes, expressed enzymes, and their reactants, while taking enantiomers into account. The five missing metabolites listed in the commercial extract, but not present in the KEGG metabolome data of *P. pastoris* comprise guanidine acetic acid (C00581), sarcosine (C00213), betaine (C00719), galactose variants (C00124, C00962, C00984, C01113, C01582, C01825), and erythritol (C00503). Therefore, the 89 validated metabolites represent 100% of the target set size in the current analysis.

DS-FIA-MS/MS analysis allowed us to verify 66 of 89 metabolites, covering up to 74.2%. The missing 23 metabolites contain 13 metabolites that were not identified and 10 metabolites that were not present in the method because of the missing metabolite information in the form of pK_a_ or MS/MS fragmentation data. While the certificate of analysis lists routinely identified metabolites, there was no precise information about the concentration and no guarantee of detectable amounts given. Hence, it is unclear if the 13 nonidentified metabolites (Figure 5a) are actually present in a detectable abundance in this specific extract preparation. Still, a potential explanation could be the applied dilution factor of 1:10^3^ and the potential low intracellular metabolite pools of certain analytes, e.g., sugars and redox cofactors. While a certain dilution is necessary to avoid nonlinear responses due to ion suppression effects [77], it may promote dilution of biological relevant below the compound-specific detection limit. This is supported by the list provided in the CoA showing varying example concentrations ranging from 0.001–5 µmol L^−1^.

The identified 66 of 89 metabolites were classified to specific KEGG metabolite classes (Figure 5a), with 39.4% amino acids, followed by 24.2% nucleic acids, 15.2% sugars and sugar phosphates, 9.1% organic acids, and 6.1% nonclassified metabolites as well as vitamins and cofactors (Figure 5b).

Interestingly, the majority of metabolites (83,3%) were identified by unique mass transitions, resulting in a low number of signals (16.7%) with potentially convoluted signals (Figure 5c). Please note, that convoluted signals mainly consisted of hexose and pentose sugars and their corresponding sugar phosphates. For such metabolites, unique precursor and product ion selection in the m z^−1^ domain is unlikely to be expected, as is a general limitation of LC-free applications.

With respect to the source of MS/MS fragment spectra, the classification of the 66 identified metabolites resulted in 69.7% in-house, 22.7% literature, and 7.58% predicted mass transitions used for verification (Figure 5d). Although it seems that for most of the biologically relevant metabolites, experimental MS/MS data were available, the predictions were necessary to complete the metabolite range beyond the manually optimised parameter database.

### 2.5. Case Study

The applicability of the method was evaluated by screening three commercially available yeast extracts (YE1, YE2, YE3) of unknown composition for the 252 *S. cerevisiae* metabolites in a DS-FIA-MS/MS approach. Besides typical acceptance criteria for LC-MS such as the occurrence of metabolites (>70%) and precision of metabolite determination (<20%) in QC samples, the DS-FIA-MS/MS was first evaluated based on an additional signal quality threshold in the form of the signal-noise ratio (SN) > 5 to guarantee sufficient signal intensity for data processing and peak area integration. Of the 252 screened metabolites, 131 (YE1), 135 (YE2) and 129 (YE3) metabolites with unique mass transitions were identified in the yeast extracts (see Appendix A). Metabolites for further analysis were selected based on precision in quality control samples and precision in the single YE samples, resulting in 118 metabolites for subsequent analysis.

The filtered data, i.e., the metabolite data present in all three YEs, were evaluated by principal component analysis (PCA) for quality control evaluation and partial least squares discriminant analysis (PLSDA) for variable selection and class discrimination. The QC samples were removed from the dataset for supervised, covariance-based PLSDA modelling. A stratified double fivefold cross validation was used to calculate model performance indicators for calibration and validation. Training and test datasets were range scaled to compare metabolites relative to the biological range. The optimal number of principal components (PC) and latent variables (LV) were identified by the goodness of prediction (Q^2^X and Q^2^Y) indicator (see Appendix A). PLSDA parameters in the form of regression coefficients and VIP scores and their corresponding confidence intervals were acquired by bootstrapping (*n* = 1000).

PCA scores of the first two PCs (Figure 6a) show a tight clustering of QC samples (mixture of all three YEs) in the centre, indicating low intrabatch variability. Most of the variance in the predictor matrix can be explained by the PCA model with three PCs (see Appendix A), displayed by the goodness of fit R^2^X = 0.789. PLSDA scores of the first two LVs (Figure 6b) show similar model and component-specific goodness of fit indicators R^2^X and R^2^X_comp_ to the PCA model. The reliable class discrimination power of the model is indicated by a goodness of fit R^2^Y = 0.992 and the small difference of R^2^Y = 0.992 to Q^2^Y = 0.971.

To decipher the metabolic differences between the YEs, discriminating metabolites were acquired by variable selection in the form of nonparametric Kruskal–Wallis tests [78], class-specific PLSDA regression coefficients (β_i,class_ ≠ 0), and class-specific variable in projection (VIP) scores (VIP_i,class_ > 1) [79]. The VIP score for a given metabolite allows to evaluate predictor variables or metabolite features that best explain response or class variance. By applying the threshold, variables that contribute the most to the underlying variance in the predictor matrix are selected. This includes the variance in the predictor matrix itself as well as orthogonal variation [80]. Class discriminating metabolites were classified based on KEGG BRITEs, providing hierarchical classification for biological objects, such as metabolites.

The variable selection resulted in 40 relevant compounds with biological roles, 18 nonclassified metabolites, and 4 phytochemicals (Figure 7a). For further analysis, compounds with biological roles were selected because of their relevance in the biochemical pathways. This resulted in 16 metabolites of the peptide class, 12 of nucleic acid class, 4 hormones, 4 vitamins and cofactors, 3 organic acids, and 1 carbohydrate (Figure 7b). While amino acids and nucleic acid class indeed represent important metabolites, the high number of discriminating metabolites for each group was surprising. At this point, this clearly indicates differences in either cultivation or processing methods used for the three yeast extracts.

The analysis of the heatmap visualization (Figure 7c) shows a similar vitamin profile for YE1 and YE3, clearly discriminated by pantothenate as displayed by the class-specific VIP score (Figure 7d). Please note, YE2 displays the strongest difference for five of the six vitamins. Interestingly, based on manufacturer information, YE2 contains added vitamins, but only for Nicotinate, a discriminating higher vitamin content was found compared with YE1 and YE3. It can only be speculated if the addition of vitamins in YE2 were not sufficient to overcome biological variance compared with the nonspiked YE1 and YE3.

With regard to the peptide group of metabolites, YE2 shows a completely different metabolite profile with lower abundances for almost all amino acids. As indicated by the corresponding VIP scores and the group size, the amino acid group is the dominating discriminating metabolite set for YE2. The amino acid profile of YE2 looks complementary to YE1 and YE3. The high abundance of proteinogenic amino acids for YE3 could be a consequence of the yeast biomass processing since YE3 is an autolysate, which might indicate high proteolytic digestion during autolysis.

Similar to the peptide group metabolites, the nucleic acid group displays an opposite metabolite profile for YE2 compared with YE1 and YE3. While YE2 shows a high abundance of pyrimidine nucleotides CMP and UMP, which could indicate an efficient ribonucleic acid (RNA) restriction digest, YE3 discriminates because of a high abundance of the nucleosides adenosine and cytidine, which are follow-up products if the nucleotides are digested by a nucleotidase. Interestingly, YE2 discriminates again by displaying a high abundance of the purine nucleobases adenine and guanine obtained by follow-up digestion of nucleosides by nucleosidases. Finally, YE3 shows a high discriminating abundance of the desoxyribonucleic acid (DNA) restriction product dCMP, which might be a result of accelerated DNA restriction digest promoted by autolysis.

The overall discriminating and mostly mirrored metabolite profile of YE2 is obvious and indicates a different feedstock, cultivation, or downstream processing procedure for this YE. Strikingly, the use of cysteine as a feedstock additive for glutathione production in yeast [81,82] is clearly indicated by the discriminating abundances of these metabolites in YE2.

## 3. Discussion

Targeted tools for large-scale metabolomic studies making use of MRM with QqQ devices are limited by the need for compound-specific MS instrument parameters, e.g., MS/MS mass transitions. Unavailable or expensive single analyte standards limit manual parameter optimization, and adequate mass transition selection for the instrument used. Previous studies regarding this topic utilised sample-dependent and incremental methods for parameter determination, with a long sample run time, if typical LC is used. To cope with such throughput challenges, the overall run time needs to be decreased, going hand in hand with an increase in the number of metabolites measured, even if no analytical standard is available.

Overall, the automated collection of metabolite information in the form of string representations such as SMILES, mass transitions, and predicted pK_a_ values obtained from established databases is obligatory for the assembly of tailor-made comprehensive metabolite sets, e.g., for a specific biological system. The use of KEGG ID and the genome-scale model of the biological system of interest (e.g., SCE = *S. cerevisiae*) is a clear benefit to the classification and identification of metabolites with unique identifiers to avoid challenges originating from trivial names of compounds. Additionally, simple updates of the databases can be achieved without additional operator time for manual curation. While the benefits of automation are obvious, database coverage is of high importance and needs to be discussed. For 5676 metabolites with pK_a_ value information or the presence of acidic and basic functional groups, the ionization mode allocation is possible. Overall, the validation of predicted pK_a_ values from ChEMBL was conducted with experimental data in the form of 374 acidic and basic pK_a_ values of acids, bases, and amphoteric molecules. Sufficient alignment for robust and reliable selection of the ionization mode was provided by a general absolute pK_a_ deviation of ±1 with few outliers up to ±4 units.

Although molecule identifiers, string representations, and MS/MS predictions can be acquired for most metabolites, the gap of missing pK_a_ values or the absence of acidic and basic groups in the metabolites represents an ongoing challenge. Without the knowledge of protonation or deprotonation capabilities of a given molecule in a soft ionization process, the random assignment to the corresponding ionization mode might lead to false positive or false negative identifications. Such random allocation was not applied in this study to keep a clear focus on metabolites with a clear description and classification, but it could be an approach to circumvent the issues arising from missing pK_a_ information. Future work needs to concentrate on increasing the pK_a_ information and metabolite spectra for those metabolites, which miss a clear ionization mode identifier at the moment.

In this study, the data for mass transitions of metabolites were collected from manual in-house optimization, literature, and predicted mass spectra. The latter were validated with in-house measurements and showed good agreement with the original validation study [53], which used data from MS/MS databases. Weighted metrics WR and WP of 85% and 65% demonstrated sufficient prediction capabilities for high-intensity signals, which might be suitable for mass transition selection. Although the prediction might be prone to error to a certain extent, it seems especially evident for small response fragments, as demonstrated by the unweighted R and P of 68% and 32% as well as J of 27%. While this might be crucial for spectra matching and library search in an untargeted approach, it is not as relevant if the strongest fragment signals yield valid mass transitions for a targeted approach. Nevertheless, goodness of predictions should be evaluated or validated with data obtained from the individual mass spectrometric hardware device used in the laboratory.

The validation with a PPA extract demonstrated the applicability of the method development workflow itself. DS-FIA-MS/MS identified 74.2% of routinely identified metabolites listed in the certificate of analysis within 7 min/sample and a single dilution step of 1:10^3^. In addition, 7.6% of common metabolites had to be predicted because of the missing standard availability. As expected, nonunique mass transitions occurred mainly for sugars and sugar phosphates due to missing LC separation.

The analysis of commercially available SCE yeast extracts for technical applications was conducted with an SCE-specific method set. MRM sensitivity allowed identifying 129–135 metabolites in a column-free DS-FIA-MS/MS approach with an analysis time of 7 min/sample. Typical QC procedures were transferred to a DS-FIA-MS/MS method to guarantee the corresponding data quality. Even though the yeast extracts for technical application may intrinsically bear higher salt loadings, we were able to show the discrimination power of the method due to MRM precision, resulting in 118 intersecting metabolites with low variability for subsequent data analysis. The PLSDA model allowed us to identify the discriminating metabolites of the yeast extracts and to generate basic hypotheses about cultivation and processing procedures. This highlights the actual applicability of the large-scale DS-FIA-MS/MS method as a valuable screening tool for high-throughput applications.

The biggest advantage of the presented method over common LCMS methods is not only the much-reduced analysis time itself but its flexibility. While the analysis time of an LCMS method is limited by the LC method time, the analysis time of the DS-FIA-MS/MS method scales with the numbers of metabolites or metabolite packages to be screened. This allows time-optimised screenings for certain metabolite sets, e.g., organism(s), pathways, or molecular classes. For SCE, the automated method development resulted, in total, in 7 mass transition packages, with 3 for the positive ionization mode and 4 for the negative ionization mode, screening 252 metabolites in 7 min per sample. Timesaving has become especially evident if the selective DS-FIA-MS/MS is compared with a 22 min per sample reverse-phase UPLC-MS/MS method [83]. Nevertheless, the potential metabolite coverage with a variable analysis time displays beneficial and relevant properties for typically time and cost-intensive applications in systems biotechnology and bioprocess development.

As based on a targeted approach, the presented method requires metabolite information prior to analysis. If the analytes and their corresponding information are known a priori, the automation of method development allows the creation of metabolite set-specific methods on demand. The metabolite coverage is based on the genome models of the organisms and the availability of metabolite information such as pK_a_ values. The current capabilities potentially allow the incorporation of multiple organism genome models and the selection of certain pathways by the operator. For fully described metabolites, coverage could be reduced to screen for operator-selected metabolites, which would result in a shorter analysis time. Nevertheless, large-scale metabolite screening, automated feature detection, and subsequent method development for shorter analysis times in an automated setup are other options. While this is certainly a time-optimising procedure, it will be a trade-off between information gain and analysis time in future activities.

Using the presented semitargeted DS-FIA-MS/MS method for targeted analysis may allow for absolute quantification of the metabolites, as demonstrated previously for amino acids. However, the availability of calibration reference substances is a major issue in this regard. Furthermore, a targeted analysis should be conducted with isotope dilution mass spectrometry for normalization. While additional screening of labelled mass transitions is simple, obtaining fully labelled metabolome standards at a low cost for a high-throughput application may be difficult. In general, the method could be applicable for different sample types such as urine and plasma samples. However, this would require a revalidation study.

Overall, the presented methodology is an alternative to column-based time-intensive semitargeted MS/MS analysis. The constantly increasing throughput of automated cultivation platforms requires a dynamical screening method for relevant targets or biological patterns. With regard to automation, the scalable DS-FIA-MS/MS avoids the problem of column or mobile phase selection and allows the creation of flexible metabolite screening methods based on operator choices or maybe even interesting features in a feedback loop. The presented methodology provides metabolite screening in a time-optimised way by scaling analysis time to metabolite coverage and is open to other microbial systems simply by starting using genome-scale model information.

## 4. Materials and Methods

### 4.1. Materials

Single metabolite standards were purchased from Sigma-Aldrich (Schnelldorf, Germany). The metabolite yeast extract standard (*Pichia pastoris*, PPA, ISO1-UNL) was purchased from Cambridge Isotope Laboratories, Inc. (Tewksbury, MA, USA). All other yeast extracts, namely Yeast extract for microbiology (YE1, 92144, Batch Number BCCC6082), Yeast extract (Vitamin enriched) for microbiology (YE2, 07533, Batch Number BCCC4059), and Yeast autolysate for microbiology (YE3, 73145, Batch Number BCCB4473) were acquired from Sigma-Aldrich (Schnelldorf, Germany). UPLC/MS-grade MeOH was obtained from Biosolve BV (Valkenswaard, Netherlands). Acetic acid (Ph. Eur.) was purchased from Roth (Karlsruhe, Germany). LC-MS grade water was obtained from a Milli-Q water purification system (Merck Millipore, Burlington, MA, USA).

### 4.2. Automated Database Creation

The source code and jupyter notebook for database creation can be found in the repository. The database creation workflow is displayed in Figure 8. KEGG identifier of microorganisms and metabolite-specific PubChem SID, as well as ChEMBL identifier, were acquired with the wrapped KEGG Rest API provided in the biopython 1.76 package. SIDs were used with pubchempy 1.0.4 to access the PubChem compound identifier (CID), representing the neutrally charged form of the molecule with the corresponding string representation (SMILES). The ChEMBL identifier was used with the chembl-webresource-client 0.10.1 to access the pK_a_ values of ChEMBL. Organism-specific pathway information was gathered with the KEGG Rest API by providing the organism-specific KEGG identifier, resulting in relevant pathways, genes, enzymes, and the metabolites involved. Metabolite fragment prediction was carried out with the CFM-ID prediction tool described in [54].

### 4.3. Automated Method Development

The source code and jupyter notebook for automated method development can be found in the repository. The method development workflow is displayed in Figure 9. For automated method creation, the necessary information in the form of the organism-specific pathways, the metabolite information file, and mass transition files were parsed.

The total number of metabolites in the KEGG database were filtered based on their occurrence in the genome model of *S. cerevisiae* (SCE). This was enabled by selecting the genes of the organism, the corresponding enzymes and metabolites involved, as well as the molecular weight (MW, 30 g mol^−1^ < precursor < 1500 g mol^−1^) of the relevant compounds. Enantiomers were filtered based on molecular weight, molecular formula, and stereochemistry descriptors. The allocation of ionization mode was conducted by pK_a_ values of acidic and basic functional groups. Amphoteric molecules were allocated to both ionization modes to provide an additional degree of freedom for later mass transition selection. With regard to parsing mass transitions, the in-house and literature data were prioritised over predicted values whenever available, and parameters were read from the corresponding xlsx file provided in the repository. Predicted mass spectra were directly read from the prediction algorithm output files. The potential mass transitions from CFM-ID prediction were acquired for three collision energies (±10 V, ±20 V, ±40 V) and a minimal relative signal threshold of 5%. Missing entrance potentials were set to 10 V, and missing cell exit potentials to 4 V. Missing declustering potentials were interpolated based on linear regression of inhouse and literature declustering potentials over precursor mass. Dwell time for all metabolites was set to 50 ms. Isobaric identification was conducted with regard to ionization modes, as well as a precursor and product ion mass to charge ratios. The selection of mass transitions was based on the ionization mode, fragment intensity ranking, and isobaric convolution pattern. If a potential isobaric convolution pattern might be present for a given metabolite in the corresponding ionization mode, an alternative mass transition avoiding a convoluted signal is selected. If the potential convolution cannot be avoided, it will be selected and labelled as a convolution accordingly. The selected mass transitions were formatted to the instrument MRM mode format and equally distributed into several methods for the corresponding ionization mode. The maximum number of mass transitions per method allowed was set to 40.

### 4.4. Standard and Sample Preparation

Single unlabeled metabolite standards were prepared as 5 mM stocks in H_2_O and stored at −80 °C. The dried metabolite yeast extract standard was stored at −80 °C until processing, reconstituted in 600 µL 50% MeOH (*v/v*), and vigorously vortexed for 2 min. SCE yeast extract stocks were prepared by dissolving 10 g L^−1^ of the corresponding product in H_2_O. Quality control samples were prepared by pooling yeast extract samples. Metabolite standard and technical extracts were diluted 1:10^3^ in 6 or 12 technical replicates with 50% MeOH (*v/v*). Replicates and quality control samples were identically allocated to two V-bottom microtiter plates and subsequently sealed with self-adhesive pierceable clear zone foil for automation (391-1264, VWR International GmbH, Darmstadt, Germany).

### 4.5. Dilute and Shoot Flow-Injection Analysis Tandem Mass Spectrometry

Acquisition of accurate mass spectra was carried out with an ESI-QqToF MS (TripleTOF6600, AB Sciex, Darmstadt, Germany). For the acquisition of accurate mass spectra, unlabeled metabolite standards were directly injected into the QqToF ion source with a flow rate of 20 µL min^−1^. Product ion spectra were acquired in product ion scan mode with a collision energy ramp ranging from ±5 V to ±130 V for the positive and negative ionization mode. For direct injection, ion source voltage was set to 5.5 kV, source temperature to 0 °C, curtain gas to 30 psi, collision gas to 5 psi, and the support gases GS1/GS2 to 50 psi/20 psi.

For DS-FIA-MS/MS analysis, an Agilent 1100 system with an Agilent 1260 Infinity II Multisampler (Agilent Technologies, Waldbronn, Germany), coupled to an ESI-QqQ (API4000, AB Sciex, Darmstadt, Germany) was used. A 1 m polyetheretherketone capillary was used to connect the autosampler directly to the Turbo V ion source. Eluents for isocratic elution mode consisted of 5% acetic acid (*v/v*) and 5% MeOH (*v/v*) (Solvent A) for positive ionization mode and MeOH (Solvent B) for negative ionization mode. The first of the prepared two microtiter plates was used to perform the screening in positive ionization mode. Switching from positive to negative ionization mode was performed using a 30 min dummy method. The second of the prepared two microplates was then used to conduct the screening in negative ionization mode. The injection sequence for both modes was led by single blanks, followed by alternating QC samples and randomised sample packages with a QC every 6 samples. The injection volume was 5 µL. For the used instrument and vendor software, automated batch creation is provided in the corresponding jupyter notebook.

For DS-FIA-MS/MS in positive ionization mode, ion source voltage was set to 5.5 kV, source temperature to 650 °C, curtain gas to 25 psi, collision gas to 6 psi, and the support gases GS1/GS2 to 50 psi/80 psi. For DS-FIA-MS/MS in negative ionization mode, ion source voltage was set to −4.5 kV, source temperature to 650 °C, curtain gas to 30 psi, collision gas to 5 psi, and the support gases GS1/GS2 to 70 psi/70 psi. All gases were nitrogen. The dwell time for every mass transition in every method package with up to 40 mass transitions was 50 ms.

### 4.6. Data Processing

#### 4.6.1. Software

Instrument control and data acquisition were carried out with Analyst 1.6.3 for the QqQ and Analyst 1.7 TF for the QqToF (AB Sciex, Darmstadt, Germany). Collision energy ramps and the corresponding PI spectra were manually evaluated with PeakView 2.2 (AB Sciex, Darmstadt, Germany). Processing of extracted ion chromatograms of the MRM mode data and noise detection were automatically conducted with the MQ4 algorithm of MultiQuant 3.0.3 (AB Sciex, Darmstadt, Germany). Data processing was conducted with Python 3.7 [84] and the packages pathlib 1.0.1, pandas 1.0.3 [85], numpy 1.18.1 [86], adjusttext 0.7.3.1 [87], biopython 1.76 [88], jupyter 1.0.0 [89], matplotlib 3.3.1 [90], pubchempy 1.0.4 [72], chembl-webresource-client 0.10.1 [91], pychemometrics 0.13.6 [92], scikit-learn 0.24.1 [93], scipy 1.4.1 [94], seaborn 0.11.1 [95], and statsmodels 0.11.0 [96].

#### 4.6.2. Statistical Analysis

The source code and jupyter notebook for data evaluation can be found in the repository. For the preliminary data processing, signals were filtered by a signal/noise > 5. For intrabatch variation, a low-order nonlinear locally estimated smoothing function (LOESS) with leave-one-out cross validation for smoothing parameter determination was used [97]. Metabolites were included for statistical analysis if they had a relative standard deviation (RSD) < 20% and < 30% missing data in the QC [98].

Univariate analysis was performed with nonparametric Kruskal–Wallis omnibus test [99] with a probability of error α = 0.05. For multivariate analysis, missing sample data were mean imputed for values missing at random (MAR) and half-of-the-minimum imputed for values missing-not-at-random (MNAR) [100]. Multivariate modelling based on principal component analysis (PCA) and multiclass partial least squares discriminant analysis (PLSDA) was conducted. Model evaluation and validation were based on stratified double fivefold cross validation in a pipeline with range scaling of the test and training predictor sets to avoid data leakage.

For hyperparameter determination, goodness of fit (R^2^X or R^2^Y) and goodness of prediction (Q^2^X or Q^2^Y) model performance indicators were used. The optimal number of hyperparameters in the form of principal components (PCs) or latent variables (LVs) was selected if the goodness of prediction indicator did not increase by 5% if another PC/LC was added. Additionally, the PLSDA model was retrained with the optimal set of parameters and evaluated by bootstrap resampling (*n* = 1000) with a replacement for percentile-based confidence intervals [101,102] of class-specific variables in projection (VIP) scores [79].

## Figures and Tables

**Figure 1 metabolites-12-00257-f001:**
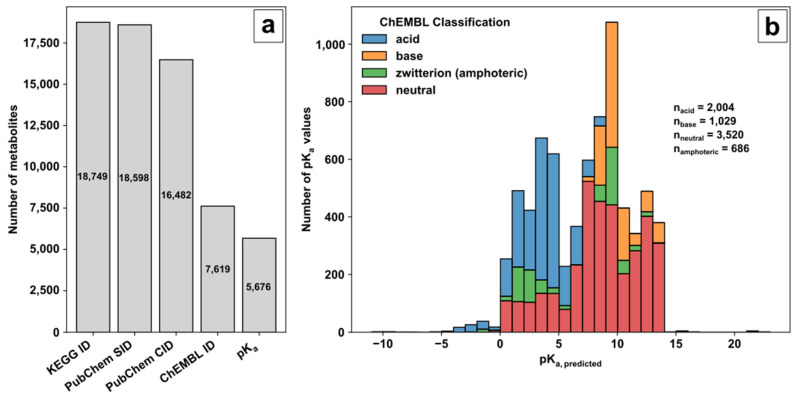
Database creation: (**a**) Filtering steps to acquire valid metabolite information for method development based on KEGG, PubChem and ChEMBL; (**b**) Histogram of pKa values with ChEMBL classifier for molecular species.

**Figure 2 metabolites-12-00257-f002:**
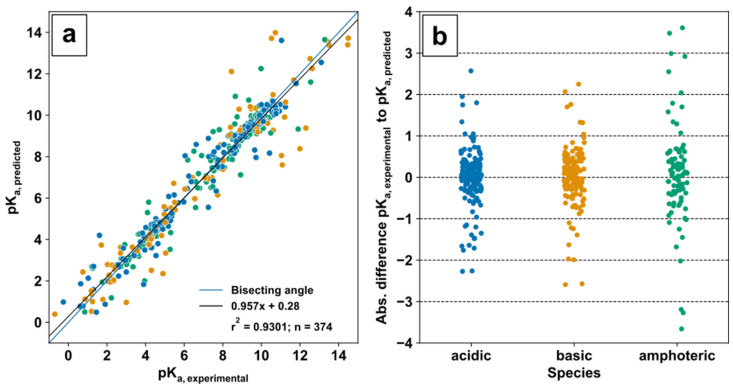
Evaluation of pK_a_ prediction for acidic, basic, and amphoteric species: Experimental pK_a_ values were obtained from [76] (**a**); Bisecting angle and linear regression of experimental and predicted pK_a_ values; (**b**) Absolute difference of predicted pK_a_ to experimental pK_a_ values.

**Figure 3 metabolites-12-00257-f003:**
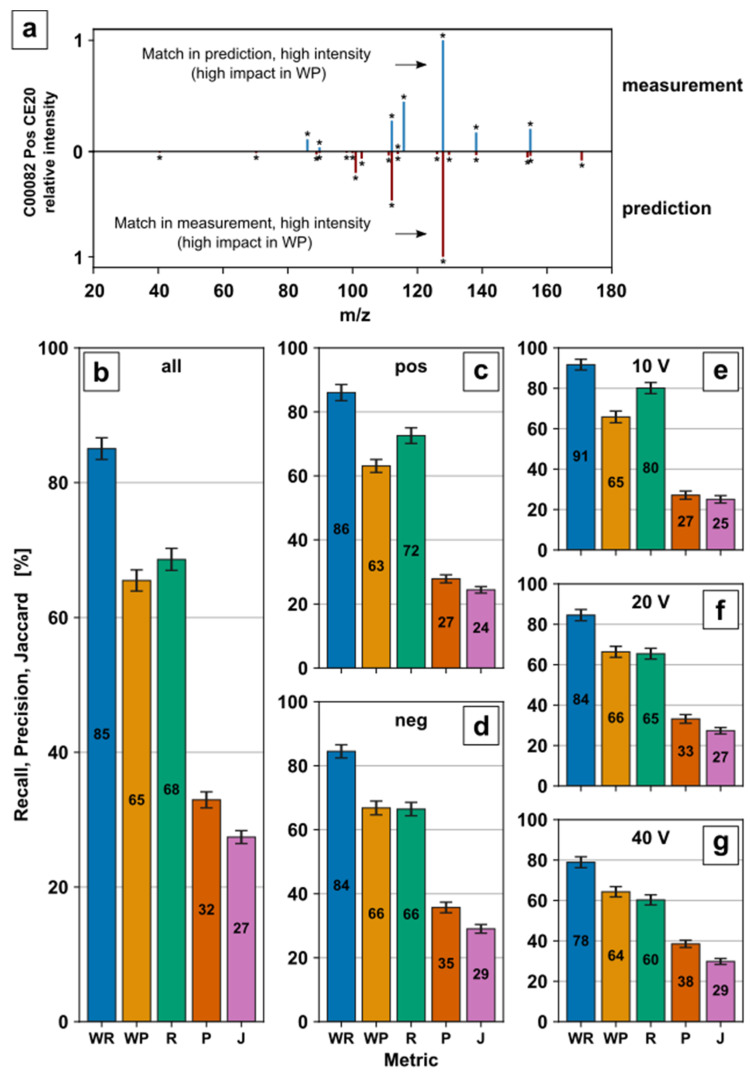
Evaluation of fragment prediction: Experimental product ion spectra of 61 metabolites were acquired by manual optimization procedure with high-resolution MS for collision energies at ±10 V, ±20 V, and ±40 V for positive and negative ionization mode (if applicable); (**a**) Measured and predicted mass spectra for tyrosine (C00082) in positive ionization mode and +20 V collision energy. An asterisk (*) highlights a fragment ion. Recall (R), Precision (P), Weighted Recall (WR), Weighted Precision (WP) and Jaccard score (J) for all predictions (**b**), predictions by ionization mode (**c**,**d**) and predictions by collision energy (**e**–**g**). Bars represent mean values with standard error.

**Figure 4 metabolites-12-00257-f004:**
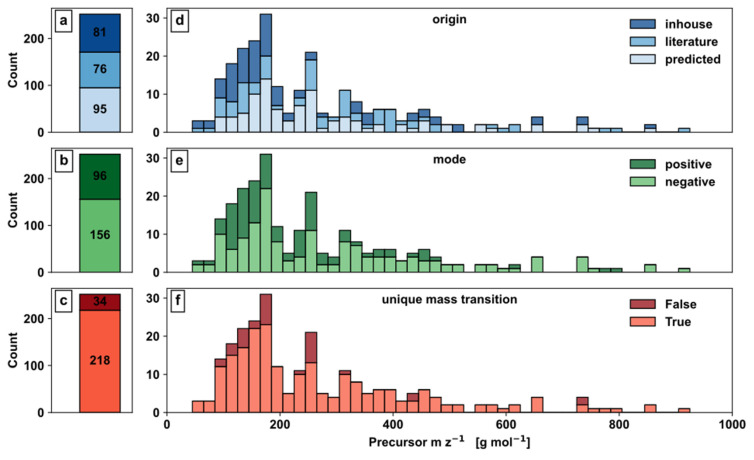
Automated method development: Barplots (**a**–**c**) and Histograms (**d**–**f**) for SCE development; (**a**,**d**) Origin information of mass transitions; (**b**,**e**) Mode information based on pK_a_ value; (**c**,**f**) Mass transition convolution.

**Figure 5 metabolites-12-00257-f005:**
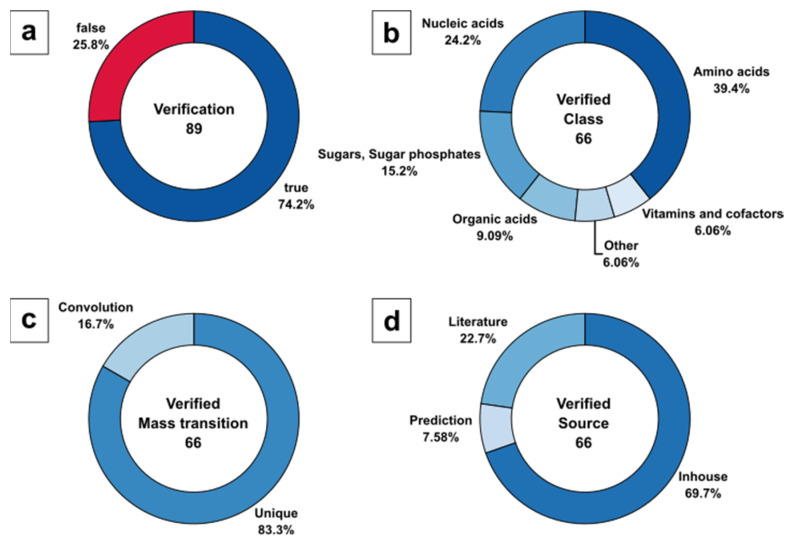
Evaluation of DS-FIA-MS/MS by metabolite verification of the PPA metabolome standard: (**a**) Identified metabolites based on certificate of analysis (routinely identified metabolites) with annotation in KEGG; (**b**) Analyte classification of verified and identified metabolites based on certificate of analysis; (**c**) Classification of verified and identified metabolites based on unique and convoluted mass transitions; (**d**) Classification of verified and identified metabolites based on mass transition source.

**Figure 6 metabolites-12-00257-f006:**
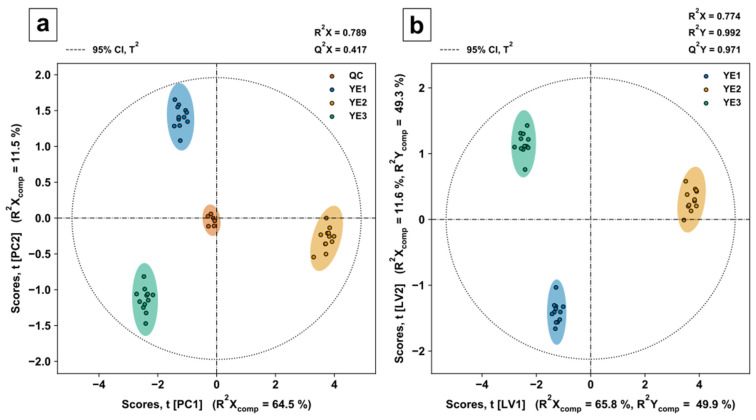
PCA and PLSDA score plots of the first and second PC or LV with Hotelling’s T^2^ ellipse for the 95% confidence interval (CI). Model performance indicators were determined by stratified double fivefold cross validation. R^2^X or R^2^X_comp_ describes the goodness of fit or explained variance of the predictor matrix by the model or corresponding PC/LV. (**a**) PCA scores plot for three yeast products (YE1, YE2, YE3) and pooled quality control samples. Q^2^X describes the goodness of prediction. (**b**) PLSDA scores plot for three yeast extract products. R^2^Y or R^2^Y_comp_ describes the goodness of fit or explained variance of the response by the model or corresponding LV. Q^2^Y describes the goodness of prediction for the response.

**Figure 7 metabolites-12-00257-f007:**
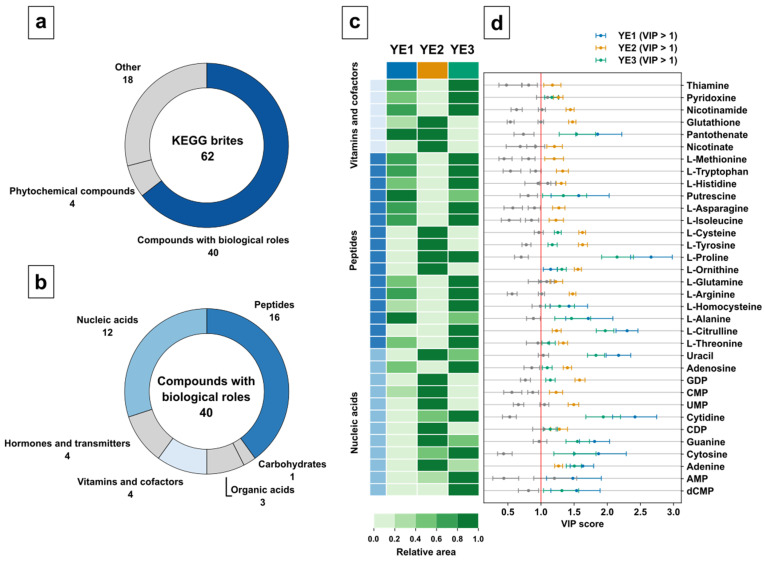
Case study evaluation by variable selection for three yeast extract products (YE1, YE2, YE3): (**a**) Top-level Categorization of discriminating metabolites based on KEGG BRITEs; (**b**) Classification of discriminating compounds with biological roles; (**c**) Heatmap by hierarchical clustering for the classes of vitamins and cofactors, peptides, and nucleic acids. The heatmap colours represent the mean relative area for each metabolite in bins; (**d**) Class-specific VIP scores for the discriminating metabolites in the vitamins and cofactors, peptides, and nucleic acids groups. VIP scores > 1 are coloured in product colours.

**Figure 8 metabolites-12-00257-f008:**
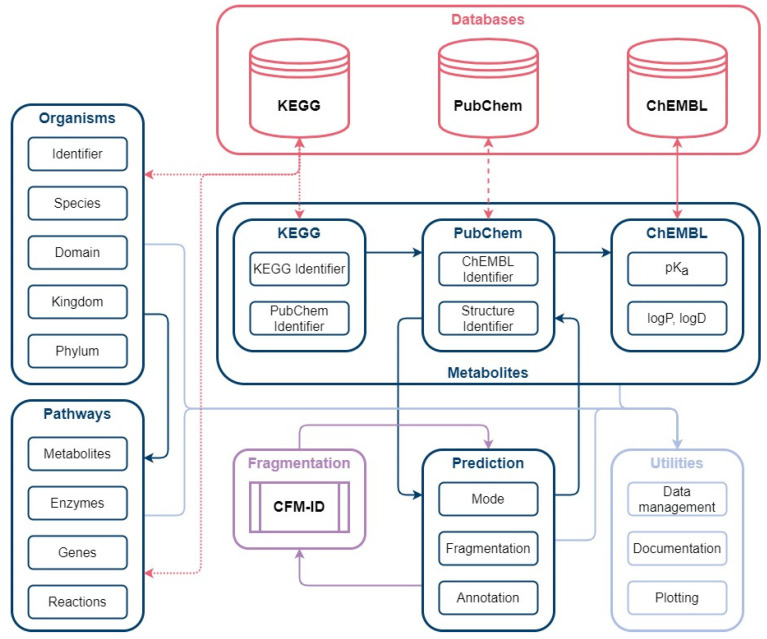
Automated database creation workflow: The workflow consists of the main modules Organisms, Pathways, Metabolites and Prediction. Organism identifiers from KEGG are used to gather organism and pathway-specific information in the form of metabolites, enzymes, genes. The Metabolites package is used to collect metabolite classification from KEGG, structure identifier from PubChem, and pK_a_ values from ChEMBL. Based on the structure identifier (SMILES), prediction of MS/MS spectra are conducted with CFM-ID.

**Figure 9 metabolites-12-00257-f009:**
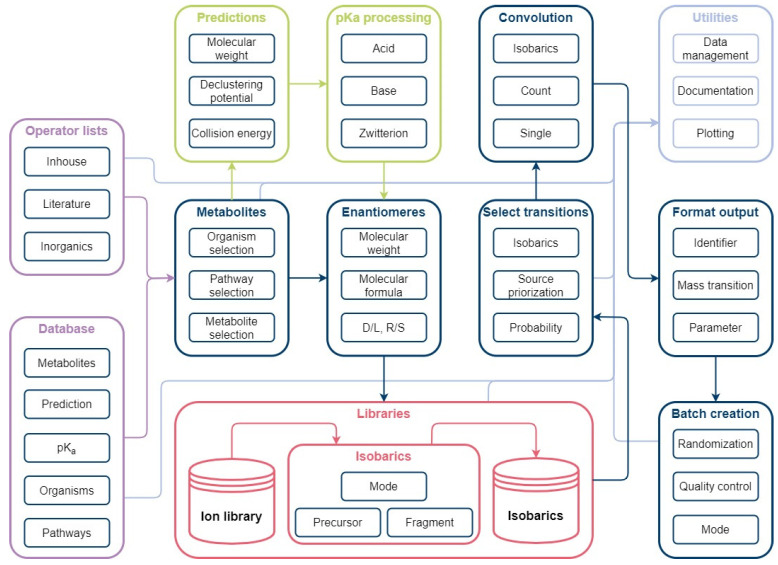
Automated method development workflow: The workflow consists of the main modules Metabolites, Enantiomers, Libraries, Select transitions, Convolution, Format output, and Batch creation. The Metabolites module is used for organism-specific selection of database information. A simple enantiomer selection is applied prior to ion library creation. Mass transitions are evaluated based on possible isobaric fragmentation patterns per ionization mode and subsequently selected in the Select transitions module. Unavoidable convolutions are considered for method creation to avoid false positives. Selected mass transitions are formatted for instrument use. Measurement lists can be generated automatically using the batch creation function.

## Data Availability

Data are contained within the article or Appendix A. Data are available in a publicly accessible repository. Accession number to the depository will be provided during review.

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
