# Peer review of "Metabolic Footprinting of Microbial Systems Based on Comprehensive In Silico Predictions of MS/MS Relevant Data"

_metabolites, 2022, doi:10.3390/metabo12030257_

Round 1

Reviewer 1 Report

The manuscript submitted by Reiter et al. is devoted to development of a method for a fast metabolome screening using HRMS/MS and QqQ MS/MS. Thу approach offered by the authors has a significant feature, namely, non-LC analysis.

The method seems to be a bit tricky in terms of composing the metabolites list and, especially, constructing the MRM method. My main questions and remarks are connected with these aspects of the manuscript. Please find these below along with some minor comments.

1) Please complete the manuscript with the references to a repository where all additional data and packets are going to be uploaded. These are references appearing as (REPOSITORY), (Link to repository).

2) No information and comments is given on the choosing the mobile phase for the screening the metabolites. Also, no data on MRM transitions for metabolites screened in the study are provided (DP, CE, Q1/Q1 etc.). To understand the method and to reproduce it, also, please provide the elution mode which allows screening metabolites in a 5 ul sample.

3) General question: Is the presented methodology applicable for either targeted or untargeted metabolomics screening of plasma/serum samples? Could the authors discuss this aspect of their approach?

Reviewer 2 Report

In this study, the author showed evidences on using in-silico predictions to obtain major parameters for targeted FIA-MS /MS method to obtain high-throughput targeted MRM method for microbial system metabolites. Overall, this study show clear experiment design, validation step and application.

  1. Please indicate you use FIA-MS/MS in abstract section, since your method is specific for FIA-MS/MS.
  2. Fragmentation pattern is also related to instrument. Does your prediction based only on QqQ instrument? You mentioned using some data from literature, are all those literature using QqQ or involve other instrument type?
  3. For figure 4, in your result section, you stated evenly distribution for figure 4b and 4e, however, the graph does not appear evenly distributed to me, may be I misunderstand your statement, please clarify.
  4. As you stated in your manuscript (figure 4c&f) that for low mass range, there're more convoluted mass transition, how confident is your method to accurately assign identity to a compound, especially for compounds from low mass range or isomers?
  5. FIA-MS/MS might be good for simple samples due to its high throughput, however, when handling complicated biological samples, ion suppression might happen which makes LC-MS/MS a better choice due to an additional clean up and separation. And RT can be used as an additional criteria for more accurate identification. Besides high throughput what is the advantage of your method compare to LC-MS/MS method?

Reviewer 3 Report

The authors developed a rapid method for mass-spectrometric classification of microbial metabolomes without using liquid-chromatography and external standards and validated their approach using yeast extracts.

Minor comments.
The authors need to discuss that inability to quantify the identified metabolites represents another cost of their rapid metabolomic footprinting approach.

In the statistical analysis, please clarify how you deal with FDR in the process of fitting between expected and observed data? 

What range of differences between the expected and observed mass of fragments did you accept?

Could you specify how you identify the level of noise in the experimental data?

Line 136 Typo in the word "ionozation"

Please, use a more recent  (i.e., 2022 year, not 2021 as in the present) template for the manuscript.
